# Isolation of Plasma Extracellular Vesicles for High-Depth Analysis of Proteomic Biomarkers in Metastatic Castration-Resistant Prostate Cancer Patients

**DOI:** 10.3390/cancers16244261

**Published:** 2024-12-21

**Authors:** Ali T. Arafa, Megan Ludwig, Onur Tuncer, Lily Kollitz, Ava Gustafson, Ella Boytim, Christine Luo, Barbara Sabal, Daniel Steinberger, Yingchun Zhao, Scott M. Dehm, Zuzan Cayci, Justin Hwang, Peter W. Villalta, Emmanuel S. Antonarakis, Justin M. Drake

**Affiliations:** 1Masonic Cancer Center, University of Minnesota, Minneapolis, MN 55455, USA; aarafa@umn.edu (A.T.A.);; 2Department of Pharmacology, University of Minnesota, Minneapolis, MN 55455, USA; 3Nuclear Medicine Division, Department of Radiology, University of Minnesota Medical School, Minneapolis, MN 55455, USA; 4Department of Medicinal Chemistry, University of Minnesota, Minneapolis, MN 55455, USA; 5Department of Laboratory Medicine and Pathology, University of Minnesota, Minneapolis, MN 55455, USA; 6Department of Urology, University of Minnesota, Minneapolis, MN 55455, USA; 7Division of Hematology/Oncology and Transplantation, Department of Medicine, University of Minnesota, Minneapolis, MN 55455, USA

**Keywords:** extracellular vesicles, proteomics, prostate cancer, B7-H3, PSMA

## Abstract

We explored the potential of using proteins found in extracellular vesicles (EVs), which are tiny particles released by cancer cells, to predict treatment response and identify new drug targets. Unlike traditional tissue biopsies, analyzing EVs from blood samples offers a non-invasive way to obtain valuable information about the cancer. Our findings showed that certain proteins in EVs, such as B7-H3 and LAT1, are correlated with clinical markers such as PSA levels, PET scans, and serum alkaline phosphatase levels that are used to monitor disease burden and cancer progression. This approach could lead to more precise and personalized treatment options for patients with advanced prostate cancer, helping doctors choose the most effective therapies and improve outcomes.

## 1. Introduction

The landscape of prostate cancer treatment has undergone a recent transformative shift with the emergence of targeted therapies. The approval of PARP inhibitors [1], checkpoint immunotherapies [2], and PSMA-targeting agents [3] has contributed to a new era of precision medicine for patients with advanced castration-resistant prostate cancer (CRPC). While these advancements offer promising therapeutic outcomes, effective patient stratification remains a critical challenge. Currently, the presence of homologous recombination repair (HHR) mutations is used to predict response to PARP inhibitors [4], while tumor mutational burden (TMB) or microsatellite instability (MSI) [2] are utilized to inform immunotherapy decision-making. For PSMA-targeting therapies, PSMA positron emission tomography (PET) scans are the primary tool to select patients for 177Lu-PSMA-617 [5]. Although these biomarkers provide valuable prognostic insights, it is necessary to develop more accurate and predictive models given the emergence of new protein-targeting therapeutics and treatment resistance. 

Currently, the 5-year survival rate for metastatic prostate cancer is approximately 50% [6]. As the disease progresses, almost all patients will eventually develop mCRPC, a stage marked by increased aggressiveness and poor prognosis, with around 90% of cases involving metastases to bone [7]. Bone metastases are a significant source of morbidity, causing severe pain, pathological fractures, spinal cord compression, and the need for palliative interventions such as radiation therapy [8]. Such complications not only diminish quality of life but also complicate treatment, further justifying the need for non-invasive biomarkers to support personalized therapeutic approaches in this large patient cohort.

The current approach to assess protein-based biomarkers has primarily relied on archival tissue analysis or invasive biopsies, both of which present significant challenges. Liquid biopsy, an emerging alternative, offers a minimally invasive approach to interrogate tumor-derived biomarkers in circulating biofluids. While circulating tumor cells (CTCs) and extracellular vesicles (EVs) have generated significant attention, the challenges associated with isolating and analyzing these biological samples have hindered their widespread clinical application [9,10]. CTCs, although informative, are rare and can be difficult to isolate and analyze. EVs, on the other hand, are more abundant and easier to isolate, making them a promising and more universal candidate for liquid biopsy-based proteomics [11].

EVs are nanosized membrane-bound vesicles originating from the inward budding of the cellular endosomal system [12]. They are actively released into the extracellular space by various cell types, including tumor cells, and carry a diverse cargo of proteins, lipids, and nucleic acids. Importantly, EVs are protected by a lipid bilayer, enabling their preservation in banked plasma samples for extended periods. This stability offers a unique opportunity to study tumor evolution and response to therapy over time [13,14,15].

Several techniques, including differential ultracentrifugation (dUC) and column-based separation, have been used for EV isolation. While these methods enrich for EVs, they often introduce contaminants that interfere with downstream proteomic analysis [16,17]. In addition, traditional nanoparticle tracking analysis (NTA) devices can detect and quantify nanoparticles, including EVs, but cannot characterize them [18]. This highlights the need for robust methodologies to isolate and characterize EVs effectively before proceeding with downstream analyses.

To address these limitations, we have developed a reproducible framework for EV characterization using mass spectrometry-based proteomics. Our approach focuses on identifying and quantifying novel EV-derived proteins that were isolated using serial dUC washes. These washes remove abundant protein contaminants found in the blood and enrich for novel biomarkers that can enable disease detection, prognosis, and therapeutic response, impacting clinical decision-making. 

## 2. Methods

### 2.1. Patient Recruitment

We prospectively enrolled 27 consecutive patients with metastatic castration-resistant prostate cancer (mCRPC) receiving systemic treatment at the Masonic Cancer Center, University of Minnesota. This study was approved by the University of Minnesota Institutional Review Board (IRB Study Number: STUDY00013815). Written informed consent was obtained from all participants prior to biofluid collection. Patients eligible for participation were those above age 18 with metastatic castration-resistant prostate cancer (mCRPC) who had previously received at least one life-prolonging systemic therapy for mCRPC (beyond ADT), who had at least one osseous metastasis, and who were willing to undergo up to 3 blood collections for biomarker analysis. There were no eligibility restrictions with respect to minimum PSA levels, minimum alkaline phosphatase, presence of visceral metastases, or presence of cancer-related pain; nor was there a cap on the number of prior systemic therapies received. Clinical information was abstracted from electronic medical records, including histologic Gleason grade, baseline PSA level, baseline alkaline phosphatase level, current and prior systemic therapies received, and the number of metastatic lesions present at the time of blood collection.

### 2.2. Mass Spectrometry

Mass spectrometry data were acquired using an Orbitrap Exploris 480 mass spectrometer coupled to a NeoVanquish nanoflow UPLC with a Nanoflex ion source (Thermo Scientific, Waltham, MA, USA) fitted with a column heater (Sonation GmbH, Model PRSO-V2, Biberach, Germany). Chromatography was performed using integrated column emitter (75 μm ID × 500 mm, 10 μm orifice) from CoAnn Technologies, LLC (Richland, WA, USA), custom-packed with Luna C18 stationary phase (5 μm particle size, 100 Å, Phenomenex Corp., Torrance, CA, USA) and maintained at 50 °C. The LC solvents were (A) 0.1% formic acid in water and (B) 0.1% formic acid in 80% acetonitrile. Peptides were separated at a flow rate of 250 nL/min using the following LC gradient: 3.8−7.5% buffer B in 2.5 min, 7.5–30% buffer B in 100 min, 30−43.8% buffer B in 10.5 min, 43.8–100% buffer B in 0.5 min, and, finally, 100% buffer B for 6.5 min at 500 nL/min, for a total run time of 120 min. The following ion source parameters were used: spray voltage = 1900 V, ion transfer temperature = 285 °C, RF lens = 45%. The data independent acquisition (DIA) tandem mass spectrometry method used the following parameters: precursor ions from 380 to 1200 m/z were scanned at a resolution of 60,000 with a maximum ion injection time of 25 ms and a normalized AGC target of 300%. The MS/MS data were acquired using HCD fragmentation at a collision energy of 28% and covering a precursor scan range from 380 to 980 m/z with 60 DIA windows centered 10 m/z apart with 1 m/z overlap and at a resolution of 15,000 with a 3 s cycle time, a normalized AGC target of 1000%, and a product ion scan range of 145–1450 m/z.

### 2.3. Processing of Mass Spectrometry-Derived Proteomic Data

The imputed proteomic data were analyzed using DIA-NN software (version 1.8.1). Thermo (Cambridge, UK). The RAW files were imported directly and converted to .dia format. The conversion was performed using the built-in tool within DIA-NN. A new spectral library was created from a FASTA file representing the canonical human proteome. The precursor ion generation included the following settings: trypsin/P digestion, allowance for up to two missed cleavages, and one variable modification. The peptide selection criteria were set to a length of 7–30 amino acids, precursor charges between 1 and 4, and precursor m/z and fragment ion m/z ranges of 300–1800 and 200–1800, respectively. The quantification matrices were generated automatically, and the precursor false discovery rate (FDR) was set to 1%.

### 2.4. Nanoparticle Tracking Analysis (NTA)

The EV samples were analyzed for size and particle concentration using ZetaView Quatt PMX-430 (Particle Metrix GmbH, Inning am Ammersee, Germany) with ZetaView Software Suite (v.1.3). The samples were diluted to manufacturer recommendations of optimal particles per view. The measurements were collected using scatter with a 488 nm laser wavelength, recording at least 9 positions with 30 s of video, a camera rate of 30 frames per second, a set temperature of 25 °C, sensitivity of 85, and a shutter speed of 200. 

### 2.5. Protein Compartmentalization

The surface proteins were first identified [19]. The subcellular compartments for the other proteins were annotated by UniProt, and when unavailable, the GO annotation (accessed by UniProt.org) was used. The compartments were singularly marked in the following hierarchy: Surfaceome > Membrane > Nucleus > Cytoplasm > other (mitochondria, secreted proteins, endoplasmic reticulum, endosome, peroxisome, golgi, and lysosome).

### 2.6. PSMA-PET Scan Imaging

The pretreatment PET/CT imaging was performed using the Siemens Biograph 64 Vision 600 digital PET/CT scanner (Malvern, PA, USA) following a minimum 1 h uptake period after administration of the radioligands, in accordance with institutional protocol. The radioligand imaging agents used were [^1^⁸F]DCFPyL (PYLARIFY^®^, Lantheus Holdings, North Billerica, MA, USA) and [⁶⁸Ga]Ga-PSMA-11 (Locametz^®^, Novartis, Basel, Switzerland; ILLUCCIX^®^, Telix Pharmaceuticals, Melbourne, Australia). The image analysis was conducted using Syngo.via software (VA20) (Siemens Healthineers, Forchheim, Germany) with anonymized patient data. For each participant, pathological uptake sites demonstrating tracer avidity across all anatomical regions (prostate, lymph nodes, bone, visceral metastasis) were identified and counted under the supervision of an experienced Nuclear Medicine physician. The number of pathological uptake sites was stratified into two categories for statistical analysis: <10 and ≥10 lesions.

### 2.7. Statistical Analysis

Mann–Whitney *t*-tests were performed for comparisons between two independent groups. For correlation analyses, non-parametric Spearman’s correlation coefficients were calculated to assess the strength and direction of associations between variables. All statistical analyses were performed using PRISM GraphPad Version 10.3.1. The results were considered statistically significant at a *p*-value threshold of ≤0.05. The significance levels were indicated as follows: * *p* ≤ 0.05, ** *p* ≤ 0.01, *** *p* ≤ 0.001, and **** *p* ≤ 0.0001. 

## 3. Results 

### 3.1. Plasma Extracellular Vesicle Isolation and Characterization from Metastatic Castration Resistant Prostate Cancer Patients

We prospectively isolated extracellular vesicles from 27 consecutive men with metastatic castration-resistant prostate cancer treated in our clinics. The median age at enrollment was 74 years (range: 44–94 years). At the time of blood draw, the median PSA level was 70 ng/mL (range: 0.5–1000 ng/mL). In terms of metastatic distribution, bone metastases were observed in all 27 patients (100.0%), lymph node involvement was noted in 17 patients (63.0%), and visceral disease was present in 6 patients (22.2%). Previous local therapies included radical prostatectomy (RP) in 7 patients (25.9%) and primary radiation therapy in 4 patients (14.8%). All patients had received leuprolide (100.0%). Other systemic therapies included abiraterone in 19 patients (70.4%), enzalutamide in 17 patients (63.0%), apalutamide in 3 patients (11.1%), and darolutamide in 2 patients (7.4%). Chemotherapy with docetaxel was administered to 21 patients (77.8%), and cabazitaxel was given to 6 patients (22.2%). Additionally, 4 patients (14.8%) had received carboplatin, while 1 patient each (3.7%) had been treated with radium-223 and olaparib. Other baseline clinical characteristics are summarized in Table 1. 

For plasma EV isolation, blood samples were collected in EDTA tubes and then centrifuged at 2000× *g* for 15 min to isolate the plasma. The plasma was then centrifuged again at 2000× *g* for an additional 10 min to pellet any remaining cells. The plasma samples were then stored at −80 °C until further use (Figure 1A) [20].

To isolate EVs, 3 mL of plasma was thawed and then centrifuged at 10,000× *g* for 30 min to remove cellular debris and apoptotic bodies (Figure 1B) [21]. Then, the plasma was transferred into a Beckman Coulter 30 mL tube, diluted in 24 mL of PBS, and ultracentrifuged at 100,000× *g* for 2 h to pellet EVs using Beckman Coulter Optima XPN-100 Ultracentrifuge SW-32 rotor (Brea, CA, USA). Once the EVs were pelleted, they were washed with 26 mL of PBS and then centrifuged at 100,000× *g* for 1 h and 20 min. The wash/centrifugation step was then repeated 3 times to purify the EVs (a total of 4 wash cycles) (Figure 1C) [22]. All steps were performed at 4 °C. Once the EVs were purified, PBS was evaporated using a speed vacuum and EVs were resuspended in 55 µL of 2% SDS containing 1X protease and phosphatase inhibitors. The protein concentration was measured using the BCA Assay (ThermoFisher, Norristown, PA, USA) according to the manufacturer’s recommendations. Once the samples were quantified, 10 mmol/L TCEP and 25 mmol/L chloroacetamide were added to the samples and incubated for 30 min at 37 °C, followed by room temperature for 15 min to reduce and alkylate cysteines. Single-pot, solid-phase-enhanced sample-preparation was then performed to remove detergents from the lysed EVs [23]. To digest the EV proteins, Lys-C and trypsin were added at 1:50 enzyme/protein ratio overnight at 37 °C. Colormetric peptide quantification (ThermoFisher) was then performed, and 700 ng of peptides were prepared for downstream mass spectrometry analysis (Figure 1D).

EVs and their corresponding proteins were isolated for mass spectrometry-based proteomic analyses according to Figure 1. Nanotracker analysis was conducted on plasma samples from 7 randomly selected patients to assess the median size and concentration of EVs. Figure 2A shows a representative EV particle size distribution from one patient sample. The average median size across the 7 samples was 136.0 nm (Figure 2B), while the average concentration was 1.43 × 10^10^ particles/mL (Figure 2C). 

To confirm EV enrichment, we examined the mass spectrometry-based quantitative levels of seven common EV proteins. Our results showed that CD9 had the highest expression among all EV markers and other proteins were detected across the patient cohort, including ALIX, FLOT1, SDCBP, CD63, TSG101, CD81 (Figure 2D, Appendix A). Interestingly, the proteins identified from the EVs revealed a diverse distribution of proteins across various cellular compartments. The surfaceome constituted approximately 11.6% of the total EV protein content, representing proteins located on the external surface of the cell membrane. Membrane-associated proteins, not present on the external surface, accounted for about 19.2% of the EV proteome. The cytoplasmic compartment comprised the largest portion at approximately 25.3%. Nuclear proteins made up 19.5% of the total EV proteome. Lastly, 24.4% of the proteins originated from other compartments, including mitochondria, secreted proteins, endoplasmic reticulum, endosomes, peroxisomes, Golgi apparatus, and lysosomes (Figure 2E, Appendix A). Overall, these findings highlight the robustness of our EV isolation method and the effectiveness of mass spectrometry for identifying and quantifying key EV-derived proteins. 

### 3.2. B7-H3 and LAT1 Proteins Correlate with Serum PSA Levels

Given that serum PSA remains the standard biomarker used in clinical practice for the diagnosis of prostate cancer and treatment monitoring in mCRPC patients, we sought to determine if EV-based proteomics identified proteins that correlated with serum PSA levels (Figure 3). Intriguingly, the highest correlations to serum PSA were observed with SLC29A1 [r = 0.74 (0.48–0.87) *p* < 0.0001] (Figure 3F), LAT1 [r = 0.60 (0.27–0.80) *p* < 0.01] (Figure 3E), and B7-H3 [r = 0.60 (0.28–0.80) *p* < 0.0001] (Figure 3B). A significant positive correlation with serum PSA levels was also observed for PD-L1 [r = 0.42 (0.04–0.70) *p* < 0.05] (Figure 3C) and MUC1 [r = 0.55 (0.21–0.78) *p* < 0.01] (Figure 3G). Surprisingly, PSMA [r = 0.29 (0.19–0.61) *p* > 0.05] (Figure 3A), STEAP1 [r = 0.04 (−0.35–0.43) *p* > 0.05] (Figure 3D), STEAP2 [r = −0.09 (−0.46–0.31) *p* > 0.05] (Figure 3E), and TROP2 [r = 0.21 (−0.20–0.55) *p* > 0.05] (Figure 3C) protein expression did not show statistically significant correlation with serum PSA levels in this patient cohort.

### 3.3. EV-Derived LAT1 Protein Expression and Bone Metastatic Burden

Given that some EV-based proteins were significantly associated with serum PSA levels and that all of our patients had bone-predominant metastatic disease, we further investigated whether the number of bone lesions correlated with the expression of specific EV proteins (Figure 4). Interestingly, TROP2 (*p* = 0.05) (Figure 4C), LAT1 (*p* = 0.01) (Figure 4E), and SLC29A1 (*p* = 0.03) (Figure 4F) protein levels were significantly higher in patients with >10 bone lesions compared to those with <10 bone lesions, as detected on PSMA-PET scans. These findings suggest that the expression of these proteins was notably higher in patients with a greater number of bone lesions and may serve as potential biomarkers or drug targets in patients with extensive bone metastases. No significant differences were observed for the remaining proteins with respect to bone lesion burden.

To further validate the association between the number of bone lesions and EV protein expression, we examined the correlation between EV protein expression and serum alkaline phosphatase (ALK) levels (Figure 5), a blood-based marker used to monitor bone metastasis. Among the proteins analyzed, a strong positive correlation was seen with B7-H3 [r = 0.73 (0.47–0.87) *p* < 0.0001] (Figure 5B). LAT1 [r = 0.49 (0.12–0.74) *p* < 0.01] (Figure 5E) continued to display a significant positive correlation with serum ALK levels. No other significant correlation was observed across the other proteins described.

## 4. Discussion

To our knowledge, we are the first group to successfully combine mass spectrometry-based EV proteomics with clinical correlates in patients with mCRPC. By performing extra dUC washes to purify EVs, followed by the integration of the corresponding proteomic data with patient clinical characteristics, we have established a comprehensive framework that sheds light on potential biomarkers that may inform clinical decision-making. This unique approach may help develop targeted therapies and patient stratification tools that enhance treatment efficacy and improve patient outcomes. 

PSMA-targeted therapeutics have paved the way for the exploration of cell-surface, proteomic-based treatments. In fact, several clinical trials are actively investigating surface proteins, such as PSMA, B7-H3, and TROP2, due to their high expression in prostate cancer [3,24,25,26]. Additionally, proteins such as PD-L1, STEAP1, and STEAP2 are associated with aggressive forms of prostate cancer [27,28]. Novel candidates identified in our dataset, such as LAT1, SLC29A1, and MUC1, further expand the landscape of potential targets. However, current approaches often rely on PET imaging or archived biopsies to identify suitable protein candidates for these therapies [29]. While effective, this method may not comprehensively capture the heterogeneous and dynamic proteomic landscape necessary for optimal patient stratification. EVs, with their ability to carry a rich profile of cell-surface proteins and other molecular markers reflective of the tumor’s state, may offer a solution to this limitation [9]. By isolating and analyzing EVs from patient plasma, we can gain deeper insights into the expression of molecules such as PSMA and B7-H3 at a proteomic level, thus providing a more accurate and comprehensive means of stratifying patients less invasively. 

Our findings highlight a significant correlation between specific EV-derived proteins and clinical characteristics, particularly for LAT1, which was not previously recognized as a relevant protein in prostate cancer. LAT1 facilitates the uptake of large neutral amino acids, such as leucine, which are essential for the activation of the mTOR signaling pathway, promoting tumor growth and proliferation. Furthermore, overexpression of LAT1 has been associated with aggressive forms of mCRPC, poor prognosis, and resistance to systemic therapies [30,31]. Targeting LAT1 may thus offer a promising therapeutic approach, particularly in advanced and treatment-resistant prostate cancer cases. The implications for clinical practice are significant, as they offer a pathway to shift from generalized treatment strategies to more personalized approaches, potentially avoiding the administration of non-efficacious treatments and enhancing patient outcomes through precision medicine.

B7-H3, another prostate cancer-relevant protein that was robustly detected in our assay, may play a central role in immune modulation and tumor progression. As a potential immune checkpoint protein, B7-H3 may promote tumor immune evasion by inhibiting T-cell activity, particularly by impairing CD8+ T-cell responses and enhancing regulatory T-cell (Treg) activity [32]. Beyond immune evasion, B7-H3 has also been implicated in angiogenesis and metastasis, making it a promising therapeutic target in mCRPC and other advanced solid tumors [33,34,35,36]. In the current study, plasma EV-derived B7-H3 protein expression was positively correlated with serum PSA and alkaline phosphatase levels and showed a trend towards a higher bone-metastatic burden. Currently, patient selection for B7-H3-targeted therapies, including antibody-drug conjugates (ADCs), relies heavily on tissue-based biopsies and immunohistochemical (IHC) analyses. Furthermore, despite the rising interest in non-invasive imaging techniques for biomarker-guided therapy, no PET imaging tracers reactive to B7-H3 are currently available, creating a critical gap in the field of molecular imaging [37]. Further, by detecting immune-regulatory proteins in EVs, our assays may capture inflammatory processes that may also contribute to progression in the presence of other viral infections [38]. Altogether, detection of B7-H3 in plasma EVs offers a highly attractive, minimally invasive platform for biomarker-guided therapy, enabling improved patient stratification and therapeutic monitoring.

In addition, both LAT1 and B7-H3 present opportunities for integration into combination therapies. For B7-H3, its immunomodulatory role suggests potential synergy with immune checkpoint inhibitors, such as anti-PD-1 or anti-CTLA4 therapies, to overcome immune resistance and enhance anti-tumor immunity [39]. LAT1, by contrast, could be targeted to disrupt cancer cell metabolism, complementing systemic treatments to improve efficacy in metabolically active tumors [40]. Moving forward, larger prospective studies will be needed to validate the roles of these biomarkers, including their predictive value for therapeutic outcomes and their potential to guide treatment decisions. Longitudinal studies using plasma-derived EVs could further elucidate the dynamic changes in B7-H3 and LAT1 expression during specific systemic treatments, providing a foundation for biomarker-driven personalized therapeutic strategies in mCRPC.

## 5. Limitations

Our study has several shortcomings. One significant limitation is the absence of clinical outcomes data from these patients, which prevents us from correlating EV proteomic findings with progression-free survival or overall survival. Future studies that include longitudinal patient follow-up and survival analysis will be essential to validate the clinical impact of these proteomic markers. Additionally, the size distribution of the EVs analyzed in our study was heterogeneous, as demonstrated through nanoparticle tracking analysis. This heterogeneity adds complexity to interpreting the clinical relevance of our findings, as different EV subpopulations may carry distinct cargos with variable biological functions. The impact of this size variability on protein identification and clinical outcomes remains incompletely understood. Further investigation into the role of specific EV subpopulations is necessary to refine the utility of this assay, and as such, this study falls short of establishing new biomarkers for cancer detection, disease progression, or disease burden. Moreover, proteins such as B7-H3 and LAT1, while robustly detected in our assay, remain unvalidated as a complement to imaging-based metastasis detection. Future studies are needed to assess their potential in improving diagnostic accuracy and clinical decision-making. We also acknowledge that we did not validate our proteomic findings with ELISA (enzyme-linked immunosorbent assay) methods, highlighting the need for additional validation studies to confirm the robustness and reproducibility of these results. Lastly, the lack of plasma EV proteomic data from non-cancer control individuals prevents us from drawing definitive conclusions about whether the detected proteins are tumor-derived or not, again falling short of defining a new cancer biomarker. Future studies incorporating EV proteomic analysis from healthy controls or individuals with benign conditions will be critical to establish the specificity of these proteins from cancer-derived EVs and for establishing them as cancer-relevant biomarkers.

## 6. Conclusions

Our study nominates plasma EV-based proteomics as a potential approach for protein-based discovery in mCRPC patients. The associations of B7-H3 and LAT1 with clinical characteristics are intriguing but require further validation in larger mCRPC cohorts, and in comparison to healthy and benign controls, with careful consideration of covariates that may impact EV profiles. Future studies should focus on confirming the clinical relevance of these proteins and exploring their integration with established diagnostic tools such as PSMA-PET and CT scans to improve patient stratification and treatment selection. Additionally, while liquid biopsy approaches have been underexplored in localized disease or scenarios with lower metastatic burden, the role of EV proteomics in earlier cancer stages remains to be determined. Addressing current limitations, such as the absence of clinical outcomes data, heterogeneity in EV size, and the lack of non-cancer control comparisons, will be critical to translating these findings into clinical practice moving forward.

## Figures and Tables

**Figure 1 cancers-16-04261-f001:**
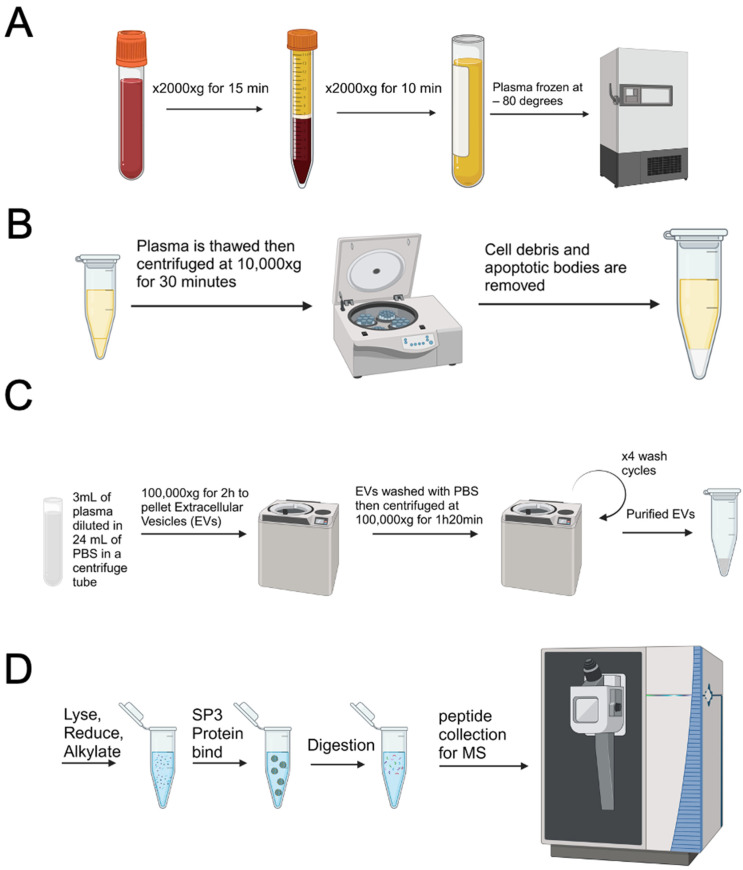
Plasma extracellular vesicle isolation workflow. (**A**) Plasma isolation from whole blood. (**B**) Centrifugation step to remove cellular debris and apoptotic bodies. (**C**) Extracellular vesicles isolation using differential ultracentrifugation. (**D**) Proteomic preparation for mass spectrometry.

**Figure 2 cancers-16-04261-f002:**
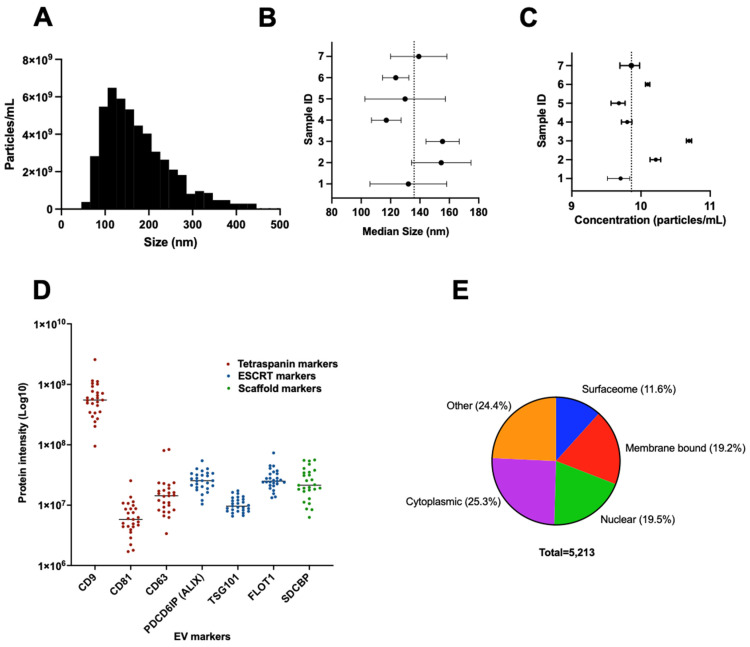
Detection of extracellular vesicles and proteins from plasma samples. (**A**) NTA demonstrating successful EV isolation from plasma. (**B**) Forest plot NTA showing median size (nm) detected from EVs. (**C**) Forest plot NTA showing the concentration of detected EV particles (particles/mL). (**D**) EV markers in plasma-derived EVs using mass spectrometry (n = 27). (**E**) Classification of EV-derived proteins detected based on cellular compartmentalization.

**Figure 3 cancers-16-04261-f003:**
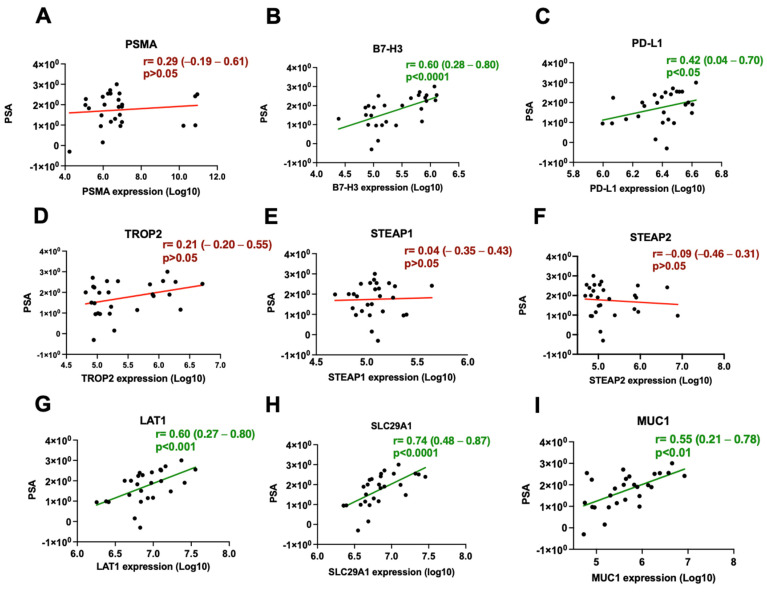
Correlations between plasma EV-derived protein expression and serum PSA levels. (**A**) PSMA, (**B**) B7-H3, (**C**) PD-L1, (**D**) TROP2, (**E**) STEAP1, (**F**) STEAP2, (**G**) LAT1, (**H**) SLC29A1, (**I**) MUC1. Significant correlations are depicted in green, while non-significant correlations are depicted in red.

**Figure 4 cancers-16-04261-f004:**
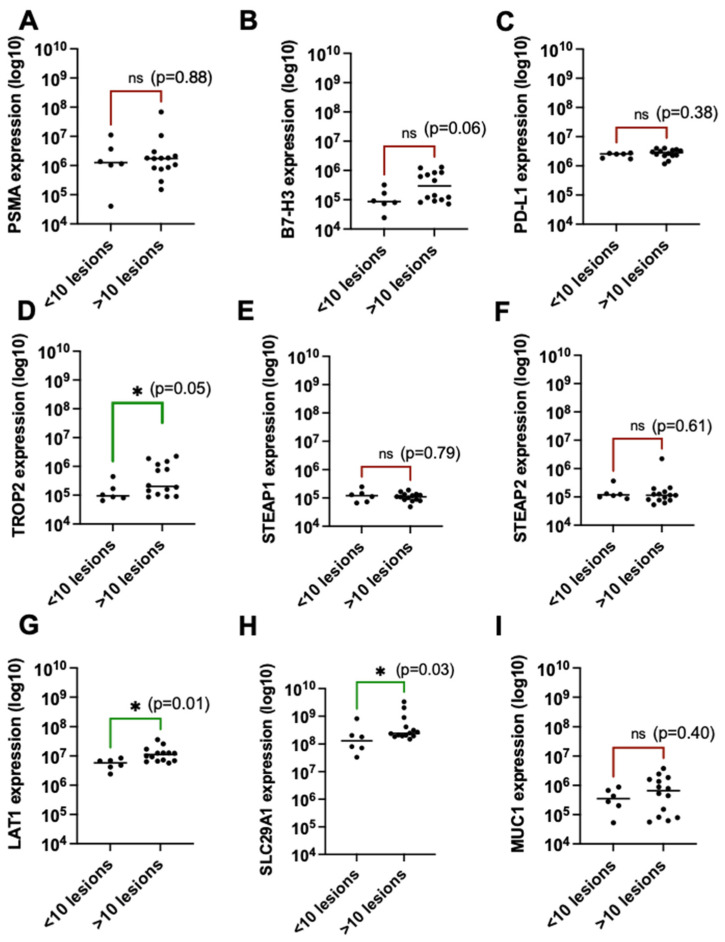
Association between plasma EV-derived protein expression and the number of metastatic bone lesions. (**A**) PSMA, (**B**) B7-H3, (**C**) PD-L1, (**D**) TROP2, (**E**) STEAP1, (**F**) STEAP2, (**G**) LAT1, (**H**) SLC29A1, (**I**) MUC1. Significant associations are depicted in green, while non-significant relationships are depicted in red. ns = nonsignificant association; * = significant association.

**Figure 5 cancers-16-04261-f005:**
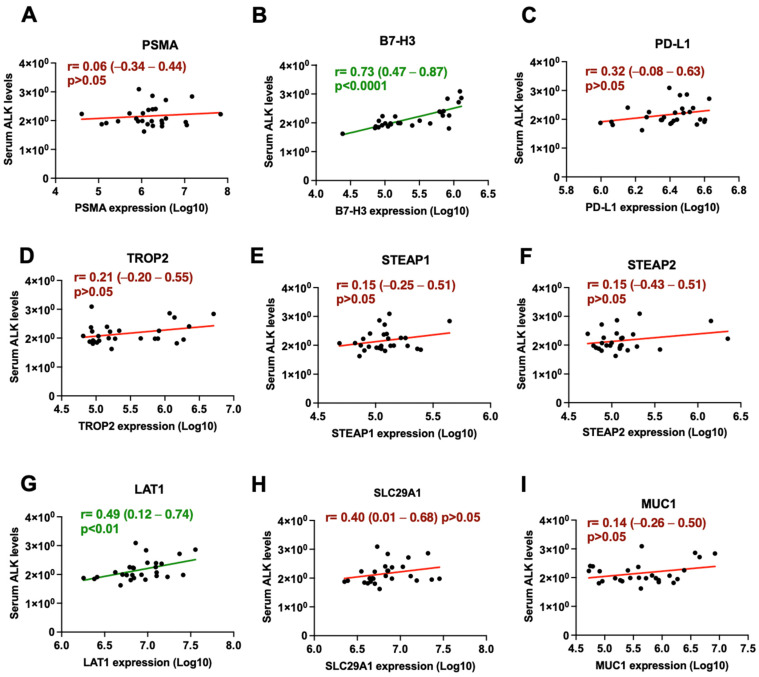
Correlations between plasma EV-derived protein expression and serum alkaline phosphatase (ALK) levels. (**A**) PSMA, (**B**) B7-H3, (**C**) PD-L1, (**D**) TROP2, (**E**) STEAP1, (**F**) STEAP2, (**G**) LAT1, (**H**) SLC29A1, (**I**) MUC1. Significant correlations are depicted in green, while non-significant correlations are depicted in red.

**Table 1 cancers-16-04261-t001:** Patient demographic and disease characteristics.

Total Number of mCRPC Patients	27
**Age at diagnosis**	
Median (min–max)	72 (42–92)
**Age at enrollment**	74 (44–94)
**PSA at diagnosis (ng/mL)**	
Median (min–max)	44 (2.6–8970)
**Race**	
White	25 (92.6)
Black	2 (7.4)
**Grade group (%)**	
1	0 (0.0)
2	2 (7.4)
3	3 (11.1)
4	5 (18.5)
5	12 (44.4)
unknown	6 (22.2)
**Baseline Serum PSA (ng/mL)**	
Median (min–max)	70 (0.5–1000)
**Baseline Serum ALK levels (U/L)**	
Median (min–max)	99 (44–1237)
**Site of disease at enrollment—no. (%)**	
Bone	27 (100.0)
Lymph node	17 (63.0)
Visceral	6 (22.2)
**Previous local therapies—no. (%)**	
RP	7 (25.9)
Radiation Therapy	4 (14.8)
None	18 (66.7)
**Number of prior systemic therapies for mCRPC**	
One	3 (11.1)
Two	8 (29.6)
Three or more	16 (59.3)
**Types of previous systemic therapies—no. (%)**	
Luteinizing Hormone Releasing Hormone (LHRH) agonist or Antagonist	27 (100.0)
**Androgen receptor pathway inhibitors (ARPIs)**	
Abiraterone	19 (70.4)
Enzalutamide	17 (63.0)
Apalutamide	3 (11.1)
Darolutamide	2 (7.4)
**Chemotherapy**	
Docetaxel	21 (77.8)
Cabazitaxel	6 (22.2)
Carboplatin	4 (14.8)
**Other**	
Radium-223	1 (3.7)
Olaparib	1 (3.7)

PSA—Prostate Specific Antigen; ALK—Alkaline Phosphatase; RP—Radical Prostatectomy.

## Data Availability

The mass spectrometry .RAW files presented in the study are openly available in ProteomeXchange.

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
