# Peer review of "Isolation of Plasma Extracellular Vesicles for High-Depth Analysis of Proteomic Biomarkers in Metastatic Castration-Resistant Prostate Cancer Patients"

_cancers, 2024, doi:10.3390/cancers16244261_

Round 1
Reviewer 1 Report
Comments and Suggestions for Authors
The article focuses on a current and relevant topic: the identification of biomarkers based on extracellular vesicles (EVs) for metastatic castration-resistant prostate cancer (mCRPC). This approach is particularly innovative, given the clinical urgency to identify non-invasive methods to improve patient stratification and personalize therapies. However, it would be useful to provide a broader overview of the clinical and epidemiological relevance of mCRPC, such as global incidence, survival rates, and the challenges in treating bone metastases. This would enhance the context and further justify the importance of the research.
The article uses a robust combination of experimental techniques, including mass spectrometry , to analyze EV-derived proteins in mCRPC patients. The correlation of proteomic data with clinical parameters (PSA, bone lesions) strengthens the validity of the study. It is necessary to provide more details about the patient selection criteria, such as age, therapies, and clinical characteristics. These data are essential to understand the applicability of the results. In my opinion, a clearer explanation of patient stratification based on protein expression would improve the transparency of the results. The description of the experimental workflow for EV isolation is adequate, but including schematic images or graphs would make the data more accessible.
The identification of biomarkers such as B7-H3 and LAT1 is promising, especially for their correlation with PSA and metastatic bone burden. These results could be relevant for the development of targeted therapies. I also suggest expanding the discussion on the molecular mechanisms through which these biomarkers influence tumor progression and therapeutic response.
The article partially addresses the role of the tumor microenvironment but does not delve into the immunological implications of EV-derived proteins. Given that biomarkers such as B7-H3 are known to modulate immune evasion, the discussion could be expanded to include their impact on immune cells such as Tregs and CD8+. For this purpose, I suggest an important study conducted on prostate cancer in patients with COVID-19 disease, which evaluates how the hyperactivated immune microenvironment from viral infection amplifies neoplastic progression:
https://pubmed.ncbi.nlm.nih.gov/32570240/
It would also be useful to propose how these proteins, B7-H3 and LAT1, could be integrated into combination treatment strategies, such as immune checkpoint inhibitors. It would be interesting to propose strategies for clinically validating the identified biomarkers, for example, through studies with a larger sample size.
The conclusions align with the data presented and highlight the potential of the identified biomarkers in improving stratification and personalization of therapies for mCRPC. However, in my opinion, it is necessary to expand the conclusions to include proposals for future research directions, such as the study of therapeutic combinations and the validation of biomarkers in clinical models. Specifically, B7-H3 and LAT1 could be used in combination with the prostate health index and multiparametric magnetic resonance imaging of the prostate to better stratify patients and their risk classes, avoiding overdiagnosis and subsequent overtreatment. For this purpose, I suggest referring to this important study:
https://pubmed.ncbi.nlm.nih.gov/35610113/
Comments on the Quality of English LanguageMinor editing
Author Response
Reviewer 1
The article focuses on a current and relevant topic: the identification of biomarkers based on extracellular vesicles (EVs) for metastatic castration-resistant prostate cancer (mCRPC). This approach is particularly innovative, given the clinical urgency to identify non-invasive methods to improve patient stratification and personalize therapies. However, it would be useful to provide a broader overview of the clinical and epidemiological relevance of mCRPC, such as global incidence, survival rates, and the challenges in treating bone metastases. This would enhance the context and further justify the importance of the research.
We appreciate the reviewer’s suggestion to provide a broader overview of the clinical and epidemiological relevance of metastatic castration-resistant prostate cancer (mCRPC). To address this, we have revised the Introduction section to include additional context on the global incidence, survival rates, and the challenges associated with treating bone metastases in mCRPC. Specifically, we have included the following:
“Currently, the 5-year survival rate for metastatic prostate cancer is approximately 50%6. As the disease progresses, almost all patients will eventually develop mCRPC, a stage marked by increased aggressiveness and poor prognosis, with around 90% of cases involving metastases to bone7. Bone metastases are a significant source of morbidity, causing severe pain, pathological fractures, spinal cord compression, and the need for palliative interventions such as radiation therapy8. Such complications not only diminish quality of life but also complicate treatment, further justifying the need for non-invasive biomarkers to support personalized therapeutic approaches in this large patient cohort.”
The article uses a robust combination of experimental techniques, including mass spectrometry, to analyze EV-derived proteins in mCRPC patients. The correlation of proteomic data with clinical parameters (PSA, bone lesions) strengthens the validity of the study. It is necessary to provide more details about the patient selection criteria, such as age, therapies, and clinical characteristics. These data are essential to understand the applicability of the results. In my opinion, a clearer explanation of patient stratification based on protein expression would improve the transparency of the results.
We appreciate the reviewer’s insightful comment regarding the need for additional details on patient selection criteria and baseline clinical characteristics. In response, we would like to draw the reviewer’s attention to Table 1 which summarizes the baseline demographic, clinical, and therapeutic characteristics of the mCRPC cohort. This includes information on age at diagnosis, age at collection, PSA at diagnosis, PSA at collection, Gleason score, number of prior systemic therapies for mCRPC, prior therapies, and sites of disease, among other factors. In addition, we have added clarifying language about our patient population: “Patients eligible for participation were those above age 18 with metastatic castration-resistant prostate cancer (mCRPC) who had received at least one life-prolonging systemic therapy for mCRPC (beyond ADT), who had at least one osseous metastasis, and who were willing to undergo up to 3 blood collections for biomarker analysis. There were no eligibility restrictions with respect to minimum PSA levels, minimum alkaline phosphatase, presence of visceral metastases, or presence of cancer-related pain; nor was there a cap on the number of prior systemic therapies received.”.
To address the reviewer’s concern about the relevance of the findings, we would like to emphasize that all patients in this study have failed at least 1 second-generation hormone therapy treatment (ADT). This distinction ensures that the proteomic analysis are comparable, enabling the identification of potential biomarkers relevant to post- second generation ADT interventions. We have made changes to the methods section to reflect this:
“We prospectively enrolled 27 consecutive patients with metastatic castration-resistant prostate cancer (mCRPC) receiving systemic treatment at the Masonic Cancer Center, University of Minnesota. This study was approved by the University of Minnesota Institutional Review Board (IRB Study Number: STUDY00013815). Written informed consent was obtained from all participants prior to biofluid collection. Patients eligible for participation were those above age 18 with metastatic castration-resistant prostate cancer (mCRPC) who had previously received at least one life-prolonging systemic therapy for mCRPC (beyond ADT), who had at least one osseous metastasis, and who were willing to undergo up to 3 blood collections for biomarker analysis. There were no eligibility restrictions with respect to minimum PSA levels, minimum alkaline phosphatase, presence of visceral metastases, or presence of cancer-related pain; nor was there a cap on the number of prior systemic therapies received. Clinical information was abstracted from electronic medical records including: histologic Gleason grade, baseline PSA level, baseline alkaline phosphatase level, current and prior systemic therapies received, and number of metastatic lesions present at the time of blood collection.”
Furthermore, we would like to highlight that this study represents a method-development effort aimed at establishing the utility of extracellular vesicle-derived proteomics in mCRPC. By correlating protein expression with baseline clinical parameters, this paper sets the stage for larger clinical studies to evaluate the impact of proteomics on prostate cancer prognosis and sensitivity to systemic therapies. The long-term goal of these efforts aim to eventually improve patient stratification and optimize therapeutic outcomes in mCRPC patients.
The description of the experimental workflow for EV isolation is adequate, but including schematic images or graphs would make the data more accessible.
We appreciate the reviewer’s suggestion to include schematic images or graphs to make the experimental workflow for EV isolation more accessible. The manuscript includes a detailed schematic in the results section (Figure 1) that visually represents the step-by-step workflow, including plasma isolation (Panel A), centrifugation to remove cellular debris and apoptotic bodies (Panel B), EV isolation using differential ultracentrifugation (Panel C), and preparation for proteomic analysis (Panel D). Additionally, the methods section provides a comprehensive and detailed explanation of each step, including how plasma is isolated, how centrifugation is performed, and the parameters used at each stage. We believe that the combination of the schematic figure and the detailed description in the text sufficiently conveys the workflow and methodology. We hope the reviewer finds these details satisfactory, but we are open to incorporating further clarifications if necessary.
The identification of biomarkers such as B7-H3 and LAT1 is promising, especially for their correlation with PSA and metastatic bone burden. These results could be relevant for the development of targeted therapies. I also suggest expanding the discussion on the molecular mechanisms through which these biomarkers influence tumor progression and therapeutic response. The article partially addresses the role of the tumor microenvironment but does not delve into the immunological implications of EV-derived proteins. Given that biomarkers such as B7-H3 are known to modulate immune evasion, the discussion could be expanded to include their impact on immune cells such as Tregs and CD8+. For this purpose, I suggest an important study conducted on prostate cancer in patients with COVID-19 disease, which evaluates how the hyperactivated immune microenvironment from viral infection amplifies neoplastic progression:
https://pubmed.ncbi.nlm.nih.gov/32570240/
It would also be useful to propose how these proteins, B7-H3 and LAT1, could be integrated into combination treatment strategies, such as immune checkpoint inhibitors. It would be interesting to propose strategies for clinically validating the identified biomarkers, for example, through studies with a larger sample size.
We thank the reviewer for these comments regarding the potential roles of B7-H3 and LAT1 in tumor progression, therapeutic response, and immune modulation. We agree that expanding the Discussion to include these aspects, as well as proposing strategies for clinical validation and combination treatments, would enrich the manuscript. In response, we have made the following revisions in the Discussion section of the manuscript and included the recommended reference:
“B7-H3, another prostate cancer-relevant protein that was robustly detected in our assay, may play a central role in immune modulation and tumor progression. As a potential immune checkpoint protein, B7-H3 may promote tumor immune evasion by inhibiting T-cell activity, particularly by impairing CD8+ T-cell responses and enhancing regulatory T-cell (Treg) activity33. Beyond immune evasion, B7-H3 has also been implicated in angiogenesis and metastasis, making it a promising therapeutic target in mCRPC and other advanced solid tumors,34,35,36. In the current study, plasma EV-derived B7-H3 protein expression was positively correlated with serum PSA and alkaline phosphatase levels, and showed a trend towards a higher bone-metastatic burden. Currently, patient selection for B7-H3-targeted therapies, including antibody-drug conjugates (ADCs), relies heavily on tissue-based biopsies and immunohistochemical (IHC) analyses. Furthermore, despite the rising interest in non-invasive imaging techniques for biomarker-guided therapy, no PET imaging tracers reactive to B7-H3 are currently available, creating a critical gap in the field of molecular imaging37. Further, in detecting immune-regulatory proteins in EVs, our assays may capture inflammatory processes that may also contribute to progression in the presence of other viral infections39. Altogether, detection of B7-H3 in circulating EVs offers a highly attractive, minimally invasive platform for biomarker-guided therapy, enabling improved patient stratification and therapeutic monitoring.
In addition, both LAT1 and B7-H3 present opportunities for integration into combination therapies. For B7-H3, its immunomodulatory role suggests potential synergy with immune checkpoint inhibitors, such as anti-PD-1 or anti-CTLA4 therapies, to overcome immune resistance and enhance anti-tumor immunity40. LAT1, by contrast, could be targeted to disrupt cancer cell metabolism, complementing systemic treatments to improve efficacy in metabolically active tumors41. Moving forward, larger prospective studies will be needed to validate the roles of these biomarkers, including their predictive value for therapeutic outcomes and their potential to guide treatment decisions. Longitudinal studies using plasma-derived EVs could further elucidate the dynamic changes in B7-H3 and LAT1 expression during specific systemic treatments, providing a foundation for biomarker-driven personalized therapeutic strategies in mCRPC.”
The conclusions align with the data presented and highlight the potential of the identified biomarkers in improving stratification and personalization of therapies for mCRPC. However, in my opinion, it is necessary to expand the conclusions to include proposals for future research directions, such as the study of therapeutic combinations and the validation of biomarkers in clinical models. Specifically, B7-H3 and LAT1 could be used in combination with the prostate health index and multiparametric magnetic resonance imaging of the prostate to better stratify patients and their risk classes, avoiding overdiagnosis and subsequent overtreatment. For this purpose, I suggest referring to this important study:
https://pubmed.ncbi.nlm.nih.gov/35610113/
We thank the reviewer for their positive feedback on the conclusions and for the valuable suggestion to expand them by including proposals for future research directions. As this study is focused strictly on the metastatic CRPC cohort, we have amended the conclusions to align with current clinical practice in the USA and Europe, which primarily utilize PSMA PET and CT scans for advanced prostate cancer patients. The roles of PHI and MRI are more applicable to localized disease. In this setting, diagnostics that capture compressive biomarkers may impact a smaller subset of patients while liquid biopsies are thought to capture substantially less tumor associated materials (Heitzer E, Perakis S, Geigl JB, Speicher MR. The potential of liquid biopsies for the early detection of cancer. NPJ Precis Oncol. 2017;1(1):36. PMID 29872715). While EVs have potential in this setting and should be explored, incorporating such statements extends beyond our conclusions in this manuscript. However, we agree that an additional clinical application of EV proteomics may be in the localized prostate cancer setting, where such biomarkers may facilitate staging and treatment decisions. In that setting, urinary (as opposed to plasma) EV proteomic biomarkers may be more relevant and our group is currently working on that approach. This will form the basis of a separate manuscript.
In response, we have revised the Conclusions section as follows: “Our research nominates plasma EV-based proteomics as a promising approach for biomarker discovery in mCRPC. The associations of B7-H3 and LAT1 with clinical characteristics suggest their potential to guide personalized therapies. Future studies should focus on validating these biomarkers in larger mCRPC cohorts and exploring their integration with other diagnostic tools such as PSMA-PET and CT scans to improve patient stratification and to aid in treatment-selection. Further, while other liquid biopsy approaches have received less consideration in the setting of localized disease or instances with lower metastatic burden, the utility of EV proteomics remains to be determined in earlier stages of disease. Finally, addressing limitations such as clinical outcomes and EV size heterogeneity will also be critical for translating these findings into clinical practice.”
Reviewer 2 Report
Comments and Suggestions for Authors
In this article the authors analyze the content of extracellular vesicles of a pull of 27 patients with metastatic prostate cancer aiming to determine novel prognostic markers that can be used to improve personalized treatment approaches.
My major concern is the fact that there is no control group. In my opinion the outcome from these patients (exosome proteomics analysis) should be compared to that of individuals with prostate cancer without metastases, and with patients with benign prostate conditions.
Since the controls are missing,
i. it is not clear whether the results obtained by the EVs analysis are associated to metastases detected by the PET scan.
ii. It is not clear whether these biomarkers expression advances the detection of metastases by imaging techniques.
I believe the authors should also compare their results with those presented in the current literature for plasma proteomics from prostate cancer patients with metastases in order to determine common biomarkers (it would be much easier to carry out a simple ELISA for a plasma biomarker rather than proceed with an additional exosome isolation and protein extraction process).
Author Response
Reviewer 2
In this article the authors analyze the content of extracellular vesicles of a pull of 27 patients with metastatic prostate cancer aiming to determine novel prognostic markers that can be used to improve personalized treatment approaches. My major concern is the fact that there is no control group. In my opinion the outcome from these patients (exosome proteomics analysis) should be compared to that of individuals with prostate cancer without metastases, and with patients with benign prostate conditions.
Since the controls are missing,
- it is not clear whether the results obtained by the EVs analysis are associated to metastases detected by the PET scan.
We appreciate the reviewer’s thoughtful comment regarding the lack of a control group and the suggestion to include comparisons with non-metastatic prostate cancer patients and individuals with benign prostate conditions. While the inclusion of a control group would indeed provide valuable insights on prostate cancer detection and/or detection of metastases, this study was specifically designed to focus on a cohort of patients who were already known to have metastatic castration-resistant prostate cancer (mCRPC) in order to identify biomarkers relevant to advanced disease stages and potential therapy-selection strategies. To address the reviewer’s concern about the specificity of the biomarkers of metastases, cell surface PSMA expression is highly specific to prostate cancer cells, with some activity also in the salivary gland, with limited expression otherwise. Therefore, PSMA act as an ideal theranostic target for metastatic prostate cancer rather than a cancer-detection biomarker in those not known to have cancer already. Prior studies have also indicated that B7-H3 is up-regulated in prostate cancer (Guo C, Figueiredo I, Gurel B, Neeb A, Seed G, Crespo M, et al. B7-H3 as a Therapeutic Target in Advanced Prostate Cancer. Eur Urol. 2023 Mar;83(3):224-38. PMID: 36114082) and further up-regulated in metastatic disease after systemic treatments (Shi X, Day A, Bergom HE, Tape S, Baca SC, Sychev ZE, et al. Integrative molecular analyses define correlates of high B7-H3 expression in metastatic castrate-resistant prostate cancer. NPJ Precis Oncol. 2022 Nov 2;6(1):80. PMID: 36323882). Lastly, we want to highlight that previous studies have performed proteomic analyses on extracellular vesicles (EVs) derived from healthy individuals as well as patients with localized prostate cancer. Notably, these studies, including the work by Welton et al. (PMID: 35340435), did not identify PSMA, B7-H3 or LAT1 as key EV-associated proteins in those settings. This supports the hypothesis that certain proteins that we found here are strongly associated with advanced or metastatic cancer.
Additionally, while we acknowledge that the current study does not include non-metastatic prostate cancer or benign prostate disease controls, we want to emphasize that the significant associations between B7-H3 and LAT1 plus the clinical features of mCRPC, such as PSA levels and metastatic bone burden detected by PET scans, strongly suggest their relevance in metastatic disease.
Another disease state where EV proteomics may be relevant is the localized prostate cancer scenario, as discussed in the last comment by Reviewer 1. In that setting, such biomarkers may facilitate staging and treatment decisions, but urinary (as opposed to plasma) EV proteomic biomarkers may be more relevant here. To this end, our group is currently working on a urine EV proteomics approach that will form the basis of a separate manuscript.
- It is not clear whether these biomarkers expression advances the detection of metastases by imaging techniques.
We thank the reviewer for raising this important point regarding the potential role of these biomarkers in advancing the detection of metastases by imaging techniques. We would like to clarify that this study is primarily a method-development effort to optimize a novel procedure to isolate extracellular vesicles (EVs) derived from the blood of metastatic CRPC patients. The proteomic findings are not ready as a suite of cancer-detection biomarkers yet, although we have captured tumor-specific proteins including PSMA and B7-H3. The efforts to apply this requires future advancements, and indeed should account for comparisons against imaging techniques for metastatic prostate cancer patients. This forms the basis of ongoing and future work by our group.
We therefore acknowledge that these EV-derived biomarkers are in the developmental phase for cancer detection and that future investigations should make comparisons against imaging techniques in cohorts of metastatic prostate cancer patients. However, we believe that the aggregate findings from our mCRPC patients set the stage for future research aimed at integrating molecular and imaging data to refine metastatic detection and guide advanced therapeutic strategies.
To clarify this, we added a statement in the Limitations section.
“Further investigation into the role of specific EV subpopulations is necessary to refine their utility as liquid biopsy biomarkers for cancer detection, monitoring disease progression, or measuring disease burden. Moreover, biomarkers such as B7-H3 and LAT1, which were robustly detected in our assay, remain unvalidated as a complement to imaging-based metastasis detection. Future studies are needed to assess their potential in improving diagnostic accuracy and clinical decision-making.”
I believe the authors should also compare their results with those presented in the current literature for plasma proteomics from prostate cancer patients with metastases in order to determine common biomarkers (it would be much easier to carry out a simple ELISA for a plasma biomarker rather than proceed with an additional exosome isolation and protein extraction process).
We thank the reviewer for this comment regarding the comparison of our findings with existing plasma proteomics studies in metastatic prostate cancer (mPCa) patients. While some plasma proteomics studies have identified biomarkers associated with metastases, such as PSA and alkaline phosphatase, our approach highlights the unique potential of extracellular vesicle (EV)-derived proteins to uncover new biomarkers that may be overlooked in plasma alone. ELISA assays are indeed an optimal panel-based approach to capture the relative expression of certain markers. However, this proteomics-based platform allowed us to unbiasedly capture up to ~5,000 proteins in plasma samples in which we have limited understanding of the molecular profiles in EVs, which may be heterogeneous across patients. While the reviewer is correct that ELISA may allow us to capture proteins through an easier process, the protocols we developed towards EV-based proteomics offers unique advantages by “enriching” for tumor-specific cargo. In addition to identifying proteins at scale (instead of a panel), this approach is not dependent on the quality of antibodies used in an ELISA assay. This is perhaps why proteins like B7-H3 and TROP2 (identified in this study) were not reported in previous plasma proteomics studies due to the difficult nature of isolating and quantifying these proteins from plasma, further justifying the added value of EV-based proteomics (Bernardino RMM, Leão R, Henrique R, Pinheiro LC, Kumar P, Suravajhala P, et al. Extracellular Vesicle Proteome in Prostate Cancer: A Comparative Analysis of Mass Spectrometry Studies. Int J Mol Sci. 2021 Dec 19;22(24):13605. PMID: 34948404). Therefore, EV-associated proteins may provide distinct insights into tumor biology and therapeutic response. We of course agree this would complement existing plasma biomarker strategies.
“Our study has several shortcomings. One significant limitation is the absence of clinical outcomes data from these patients, which prevents us from correlating EV proteomic findings with progression-free survival or overall survival. Future studies that include longitudinal patient follow-up and survival analysis will be essential to validate the clinical impact of these proteomic markers. Additionally, the size distribution of the EVs analyzed in our study was heterogeneous, as demonstrated through nanoparticle tracking analysis (NTA). This heterogeneity adds complexity to interpreting the clinical relevance of our findings, as different EV subpopulations may carry distinct cargos with variable biological functions. The impact of this size variability on protein identification and clinical outcomes remains incompletely understood. Further investigation into the role of specific EV subpopulations is necessary to refine their utility as liquid biopsy biomarkers for cancer detection, monitoring disease progression, or measuring disease burden. Moreover, biomarkers such as B7-H3 and LAT1, which were robustly detected in our assay, remain unvalidated as a complement to imaging-based metastasis detection. Future studies are needed to assess their potential in improving diagnostic accuracy and clinical decision-making. Lastly, we acknowledge that we did not validate our proteomic findings with ELISA (enzyme-linked immunosorbent assay) methods, highlighting the need for additional validation studies to confirm the robustness and reproducibility of these results.”
Round 2
Reviewer 1 Report
Comments and Suggestions for Authors
Authors answered all comments and suggestions
Author Response
Thank you!
Reviewer 2 Report
Comments and Suggestions for Authors
I am afraid, in my personal opinion, the controls are vital in order to draw some definite conclusions.
Author Response
We thank the reviewer for their valuable feedback regarding the importance of including appropriate controls to draw definitive conclusions with relation to diagnostics or using this approach to capture biomarkers. In this second revised manuscript (R2), we have altered the language that would give this impression and have made several additional updates in the Study Limitations and Conclusions sections.
In this method development study, we are able to collect EVs that contain proteins known to be expressed in CRPCs and this is correlative to disease burden. Again, we do not conclude that we have developed an approach that would be diagnostic or EV proteins that serve as a suite of biomarkers, as these would indeed require further validation with non-cancer controls, and we would need to carefully consider elements such as prior treatments, disease stage, etc.
As a method development study, our objective was to demonstrate the robustness and feasibility of our EV proteomics approach. Notably, we would like to highlight that prior studies, such as those by Welton et al. (PMID: 35340435) investigating plasma and serum from non-cancer individuals, have never captured clinically-relevant prostate cancer-associated proteins such as PSMA, B7-H3, STEAP1, and STEAP2. In contrast, our approach identified these proteins with high confidence, emphasizing the sensitivity and robustness of our assay. For example, we identified >5000 proteins compared to ~4000 proteins in prior studies, showcasing the enhanced depth and performance of our platform. PSMA is particularly noteworthy, as it is FDA-approved for imaging and therapeutic purposes in mCRPC (Sartor et al., NEJM 2021 PMID: 34161051). B7-H3 also has emerging clinical relevance in cancer, with active therapeutic development, as highlighted in recent work by Shenderov et al. Nature 2023 (PMID 37012549). Lastly, STEAP1 is also currently in clinical trials as a therapeutic target for prostate cancer (PBID: 37861461) and was detected by our assay.
To avoid any confusion in our conclusions, and in an attempt not to overstate the biomarker potential of our study, we have added more text to the Study Limitations:
“Lastly, the lack of plasma EV proteomic data from non-cancer control individuals prevents us from drawing definitive conclusions about whether the detected proteins are tumor-derived or not, again falling short of defining a new cancer biomarker. Future studies incorporating EV proteomic analysis from healthy controls or individuals with benign conditions will be critical to establish the specificity of these proteins from cancer-derived EVs and for establishing them as cancer-relevant biomarkers.”
We have also updated the Conclusions:
“Our study highlights plasma EV-based proteomics as a potential approach for protein-based discovery in mCRPC patients. The associations of B7-H3 and LAT1 with clinical characteristics are intriguing but require further validation in larger mCRPC cohorts, and in comparison to healthy and benign controls, with careful consideration of covariates that may impact EV profiles. Future studies should focus on confirming the clinical relevance of these proteins and exploring their integration with established diagnostic tools such as PSMA-PET and CT scans to improve patient stratification and treatment selection. Additionally, while liquid biopsy approaches have been less explored in localized disease or scenarios with lower metastatic burden, the role of EV proteomics in earlier cancer stages remains to be determined. Addressing current limitations, such as the absence of clinical outcomes data, heterogeneity in EV size, and the lack of non-cancer control comparisons, will be critical to translating these findings into clinical practice moving forward.”
Round 3
Reviewer 2 Report
Comments and Suggestions for Authors
Fair enough.